# Multimodal switching of a redox-active macrocycle

Daniel T. Payne [1,2], Whitney A. Webre[3], Yoshitaka Matsushita[4], Nianyong Zhu[5], Zdeněk Futera [6], Jan Labuta[1], Wipakorn Jevasuwan[1], Naoki Fukata [1], John S. Fossey [2], Francis D'Souza[3], Katsuhiko Ariga[1,7], Wolfgang Schmitt[5] & Jonathan P. Hill [1]

Molecules that can exist in multiple states with the possibility of toggling between those states based on different stimuli have potential for use in molecular switching or sensing applications. Multimodal chemical or photochemical oxidative switching of an antioxidant-substituted resorcinarene macrocycle is reported. Intramolecular charge-transfer states, involving hemiquinhydrones are probed and these interactions are used to construct an oxidation-state-coupled molecular switching manifold that reports its switch-state conformation via striking variation in its electronic absorption spectra. The coupling of two different oxidation states with two different charge-transfer states within one macrocyclic scaffold delivers up to five different optical outputs. This molecular switching manifold exploits intramolecular coupling of multiple redox active substituents within a single molecule.

[1] WPI Center for Materials Nanoarchitectonics, National Institute for Materials Science, Namiki 1-1, Tsukuba, Ibaraki 305-0044, Japan. [2] School of Chemistry, University of Birmingham, Edgbaston, Birmingham, West Midlands B15 2TT, UK. [3] Department of Chemistry, University of North Texas, 1155 Union Circle, 305070 Denton, Denton, TX 76203, USA. [4] Research Network and Facility Services Division, National Institute for Materials Science (NIMS), 1-2-1 Sengen, Tsukuba, Ibaraki 305-0047, Japan. [5] School of Chemistry, Trinity College Dublin, The University of Dublin, College Green, Dublin 2, Ireland. [6] School of Chemical & Bioprocess Engineering, University College Dublin, Belfield, Dublin 4, Ireland. [7] Department of Advanced Materials Science, Graduate School of Frontier Sciences, The University of Tokyo, 5-1-5 Kashiwanoha, Kashiwa, Chiba 277-8561, Japan. Correspondence and requests for materials should be addressed to J.P.H. (email: Jonathan.Hill@nims.go.jp)

Molecules that possess multiple stable states that are interconvertible by applying a stimulus, such as light, temperature or pH, are of interest from a chemical scientific viewpoint and because of their potential uses in molecular devices[1]. Numerous classes of molecules, including rotaxanes[2], diarylethenes[3] and luminophores[4], have been studied because of their mechanical, electronic or optical properties. In some of these cases, oxidation states or charge-transfer (C-T) interactions play roles in their operation by stabilizing individual switching states with characteristic optical or electronic states[5]. C-T interactions occur when electron density is transferred from electron-rich to electron-deficient moieties. When these moieties are respectively hydroquinone and benzoquinone, the resulting C-T complex is known as a quinhydrone[6,7] (Fig. 1a), which is characterized by an intense visible electronic absorption band. Despite much interest in C-T interactions, a corresponding interaction involving phenol-hemiquinone has not been reported (Fig. 1b) but ought also to operate according to their proximity, leading to what we term a 'hemiquinhydrone' complex.

Resorcinarenes[8,9] are macrocycles formed by cyclotetramerization of an aldehyde and resorcinol. There are four common conformers of resorcinarenes, based on the relative orientations of substituents at their meso-positions, denoted as rctt, rccc, rcct and rtct. Resorcinarenes present unique opportunities for host–guest chemistry because of their ability to form nanometric oligomacrocyclic capsules[10–13] directed by the conformational structure of the macrocyclic core. The presence of multiple hydrogen-bonding phenol groups opens opportunities to exploit self-assembly properties, conduct further synthetic elaboration[14,15] or probe their coordination chemistry[16,17]. Conversely, the effect of meso-substituent identity on the chemical properties of resorcinarene macrocycles has not been investigated in depth despite the possibility of assessing intramolecular processes based on the isomeric structures of the compounds.

In this study we investigate the coupling of an individual molecule's oxidation states with their complementary C-T states. This combination has not been reported in the context of molecular switching despite possibly providing a greater multiplicity of switchable states in a compact molecule. To achieve this, our molecular design includes multiple oxidation states and substituents that are capable of supporting discrete intramolecular C-T interactions forming the basis for a multistate optical switch operated by applying different stimuli. As a molecular scaffold, the resorcinarene macrocycle is selected because of the form and proximity of substituents at the macrocylic meso-positions. For our purposes, isomers having adjacent substituents

are required (i.e., rtct is not suitable) for the formation of intramolecular C-T interactions between meso-substituents[7,8]. We have used 3,5-di-t-butyl-4-hydroxyphenyl (DtBHP) groups[18] as meso-substituents due to their propensity to be oxidized under mild conditions allowing for the introduction of multiple oxidation states in 2. Thus, DtBHP groups have the dual purposes of introducing oxidation state variability while also leading to the presence of the requisite phenol and hemiquinone substituents required to establish a hemiquinhydrone C-T complex.

## Results

**Molecular design**. Our molecular design involves synthesis of the resorcinarene 1[19] containing multiple switchable DtBHP phenol groups in order to stabilize different oxidation states (Fig. 2a). There are two important aspects of 1: first, it can be selectively benzylated at 1,3-catechol groups[20] (2), due to the steric hindrance around the DtBHP phenol group, removing those hydroxyl groups as complicating factors and, second, the structure supports oxidation of the DtBHP groups (due to the presence of the meso-methine protons) facilitating hemiquinone formation[21]. As expected, rctt-2 could be prepared exclusively with several accessible stable oxidation states and was also found to be prone to selective photooxidation. These properties allow the chemical and molecular structure of 2 to be altered by applying ultraviolet (UV) irradiation, chemical oxidation or by stabilizing intramolecular C-T interactions, as a multimodal switching molecule.

**Synthesis**. Resorcinarene 1 was prepared selectively as a single isomer (rctt; for a graphical illustration of the other isomer forms see Fig. 2b) by the acid catalyzed condensation of resorcinol with 3,5-di-t-butyl-4-hydroxybenzaldehyde. The exclusive formation of rctt-1 is consistent with previous reports for the synthesis of resorcinarenes from aryl aldehydes[22,23]. Phenolic hydroxyl groups of the macrocycle were selectively benzylated to give rctt-2. Octabenzylate rctt-2 can be oxidized to rctt-2-[Ox$_1$] then rccc-2-[Ox$_2$] concurrently using 2,3-dichloro-5,6-dicyano-1,4-benzoquinone (DDQ) or in a stepwise manner first by photo-induced oxidation (UV 285 nm) to rctt-2-[Ox$_1$] then treatment with DDQ to rccc-2-[Ox$_2$]. For synthetic details and characterization, see Supplementary Methods and Supplementary Figures 20–22 and 26–37. Three-dimensional representations of the chemical structures of the compounds are shown in Fig. 2c with a corresponding illustration of the transformations of the compounds shown in Fig. 2d. The notable features of these transformations are the selective formations of oxidized states rctt-2-[Ox$_1$] and rccc-2-[Ox$_2$], and conformer switching during oxidation from rctt-2-[Ox$_1$] to rccc-2-[Ox$_2$], which is a seldom observed phenomenon in resorcinarene macrocycles in the absence of ring opening. rctt-2-[Ox$_1$] has its hemiquinonoid substituent delocalized over two diagonally opposing sites in its X-ray crystal structure, although in solution it is clearly localized at one site. Treatment of either rctt-2 or rctt-2-[Ox$_1$] with DDQ leads to the same isomeric product rccc-2-[Ox$_2$]. Crystal structures of all the compounds are contained in the supporting information (see also Supplementary Figures 1–9). The oxidized compounds are stable in the solid state over periods of several years. In neutral solutions, the oxidized compounds showed no observable degradation either during extended spectroscopic measurements or while under crystallization for X-ray measurements.

**Accessing different oxidation states of** 2. Electronic absorption spectra of the 2 series of compounds are shown in Fig. 3a. The absorbance at 285 nm assigned to the phenol π–π$^*$ absorption is accompanied by a new broad band at 415 nm in rctt-2-[Ox$_1$],

**Fig. 1** Formation of quinhydrone and hemiquinhydrone. **a** Quinhydrone complex formed between an electron-rich hydroquinone and electron-deficient benzoquinone. **b** Proposed formation of hemiquinhydrone complex in compounds containing 3,5-di-t-butyl-4-hydroxyphenyl groups

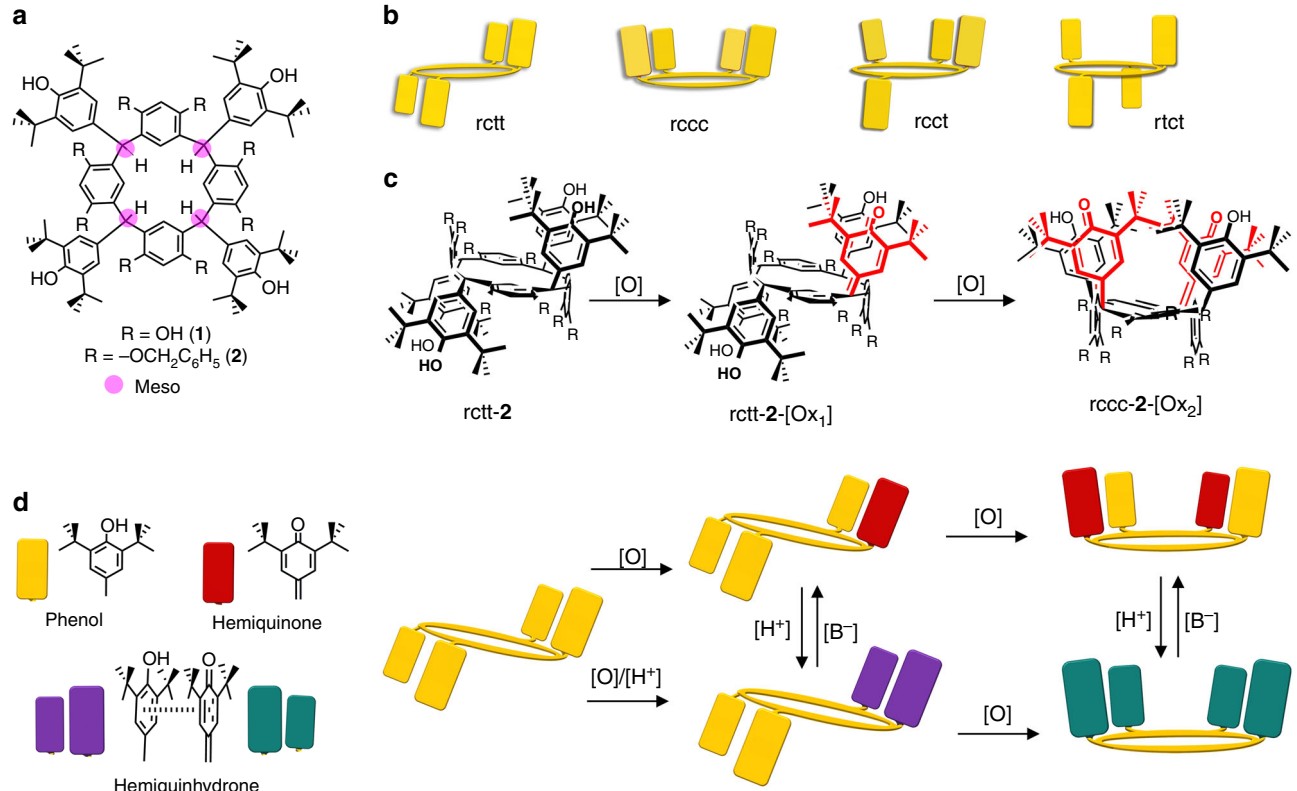

**Fig. 2** Structures and graphical representation of the compounds. **a** Chemical structures of **1** and **2** prepared in this work. **b** Common possible isomers of resorcinarene based on the relative conformations of substituents. **c** Chemical structures of compounds in the **2** series: rctt-**2**, rctt-**2**-[Ox$_1$] and rccc-**2**-[Ox$_2$] obtained by stepwise oxidation with 2,3-dichloro-5,6-dicyano-1,4-benzoquinone (DDQ). Identity of the hemiquinonoid substituents is denoted by red in the line drawings. **d** Graphical representation of transformations of **2** involving oxidation of phenol groups (yellow blocks) to hemiquinonoid (red blocks), and quinhydrone activity of the resulting compounds denoted by purple for rctt-**2**-[Ox$_1$] and turquoise for rccc-**2**-[Ox$_2$]. [O] denotes oxidation; [H$^+$] and [B$^-$] denote acid and base, respectively

which increases in relative intensity in rccc-**2**-[Ox$_2$]. The new band for rctt-**2**-[Ox$_1$] is associated with a π–π* transition containing hemiquinone conjugated with a single dialkoxybenzene unit; for rccc-**2**-[Ox$_2$], the 415 nm band is correspondingly intensified since it contains two such units. While rctt-**2**, rctt-**2**-[Ox$_1$] and rccc-**2**-[Ox$_2$] are connected by chemical oxidation processes, we were surprised to find that rctt-**2**-[Ox$_1$] could also be accessed from rctt-**2** by photochemical oxidation, although, conversely, rccc-**2**-[Ox$_2$] could not be obtained by irradiation of rctt-**2**-[Ox$_1$]. Evidence for the photochemical oxidation is shown in Fig. 3b. Irradiating a solution of rctt-**2** leads to the appearance and growth of a band at 420 nm corresponding to rctt-**2**-[Ox$_1$]. An additional band appears at 600 nm, which disappears upon neutralization of the solution with K$_2$CO$_3$ (HCl is formed by decomposition of CHCl$_3$ under UV irradiation). $^1$O$_2$ or another reactive oxygen species is generated during the reaction as revealed by appearance of a characteristic electron spin resonance (ESR) spectrum obtained by performing the photooxidation process in the presence of a spin trap[24] (see Supplementary Figure 10). rctt-**2** undergoes two oxidation processes at 1.1 and 1.3 V but these are rather poorly defined even by using differential pulsed voltammetry. rctt-**2**-[Ox$_1$] undergoes both oxidation (1.5 V) and a single reduction (−1.5 V) as expected (see Supplementary Figure 11). While the reduction (presumably to rctt-**2**) is reversible, the oxidation process was found to be poorly reversible. rccc-**2**-[Ox$_2$] also undergoes both oxidation and reduction but there appear multiple processes in each case (see Supplementary Figure 11).

**Charge-transfer switching**. Apart for the available discrete oxidation states of the **2** series, there are additional switching modalities inherent in systems containing proximal phenol and quinone moieties introduced by the possibility of the existence of C-T states. That is, rctt-**2** dissolved in acidic solution (acidified using trifluoroacetic acid (TFA)) was irradiated with UV light (285 nm) leading to the appearance of the deep purple color of its intramolecular hemiquinhydrone complex and associated C-T band at 550 nm (Fig. 3c). Neutralization of the resulting deep purple solution with pyridine gives a yellow solution with an electronic absorption spectrum characteristic of rctt-**2**-[Ox$_1$]. This also constitutes a further proof that rctt-**2**-[Ox$_1$] is formed during photoirradiation of rctt-**2**. This process can be operated in reverse by applying Zn metal in acetic acid for reduction back to rctt-**2**, albeit in low yield due to the instability of rctt-**2** in acidic media. While forward oxidation processes can be performed in either neutral or acidic media (see Supplementary Figure 12), there also exists the possibility of toggling between 'On' and 'Off' states of the C-T absorption bands. This entails consecutive treatment with acid and base as depicted in Fig. 4 and leads to reasonably reversible absorbance recovery notwithstanding some solvent loss due to evaporation during filtration. Figure 4a, b shows the results for rctt-**2**-[Ox$_1$] and rccc-**2**-[Ox$_2$], respectively. This process can also be performed in the solid state as shown in Fig. 4c. In the solid state, if a volatile acid such as TFA is employed for C-T formation, then neutralization with base is not required since evaporation of the acid returns the molecule to its original state. If we consider the different features of the **2** system overall, it is feasible to construct a molecular switching manifold based on the

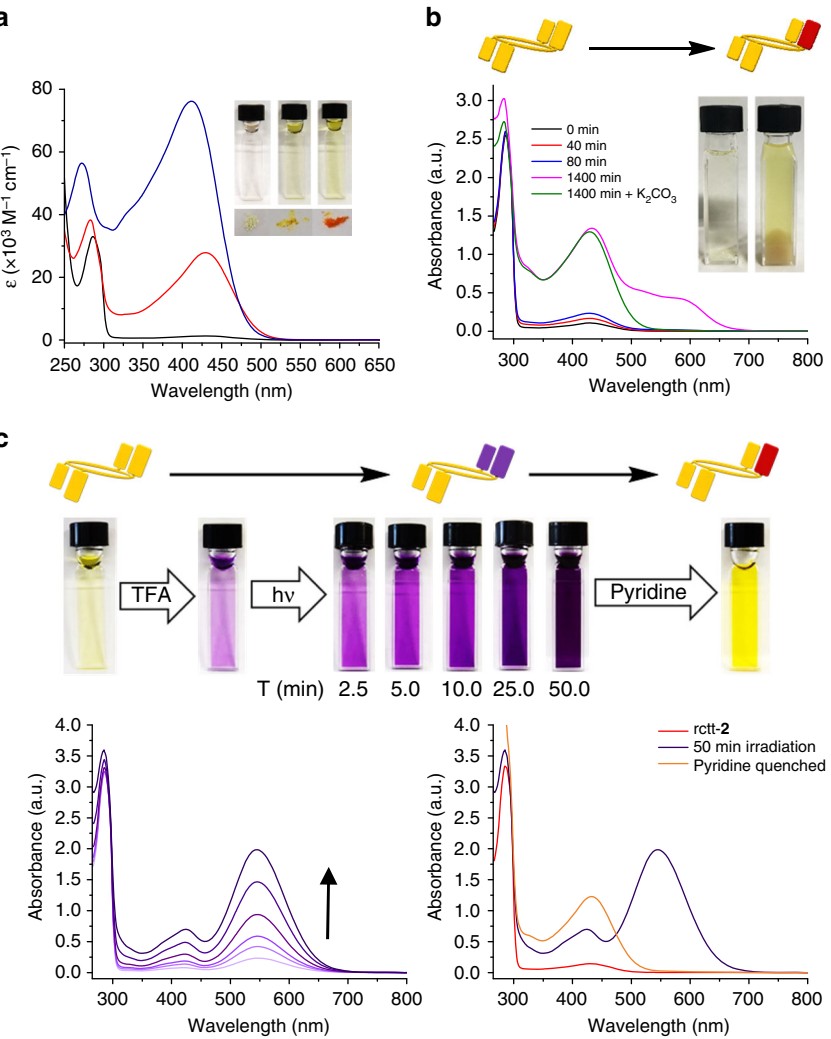

**Fig. 3** Variations in electronic absorption spectra during oxidation and hemiquinhydrone formation in **2** series compounds. **a** Electronic absorption spectra for series **2** compounds: rctt-**2** (black), rctt-**2**-[Ox$_1$] (red), rccc-**2**-[Ox$_2$] (blue) in CHCl$_3$ at $2.47 \times 10^{-5}$ mol dm$^{-3}$. Inset shows compound colors in solution (upper) and solid states (lower): (left) rctt-**2**, (middle) rctt-**2**-[Ox$_1$], (right) rccc-**2**-[Ox$_2$]. rctt-**2** is white in its native state but turns yellow in air due to the increasing presence of rctt-**2**-[Ox$_1$]. **b** Electronic absorption spectra of rctt-**2** during photoirradiation at 285 nm under ambient conditions at $9.02 \times 10^{-5}$ mol dm$^{-3}$. Increasing absorbance at 420 nm is due to formation of rctt-**2**-[Ox$_1$]. Band emerging at 600 nm is due to formation of the acid-stabilized charge-transfer (C-T) complex formed as HCl is generated in situ by the ultraviolet (UV)-promoted decomposition of CHCl$_3$. **c** Photooxidation of rctt-**2** ($1.02 \times 10^{-3}$ mol dm$^{-3}$) to rctt-**2**-[Ox$_1$] in CHCl$_3$ in the presence of trifluoroacetic acid (TFA; 10 μL, final concentration = $8.9 \times 10^{-2}$ mol dm$^{-3}$) followed by neutralization by addition of pyridine (10 μL). Photographs of cuvettes illustrate distinct color change from colorless to deep purple then to yellow upon neutralization. UV/visible (vis) spectra (bottom left) reveal emergence of a band at 550 nm and addition of pyridine yields neutral rctt-**2**-[Ox$_1$] (bottom right). Photoirradiation of rctt-**2** gives exclusively rctt-**2**-[Ox$_1$] with no higher oxidation products detectable. The initial purple hue found after addition of TFA is due to traces of rctt-**2**-[Ox$_1$] formed by photooxidation of rctt-**2** during storage

multiple interactions present. The coupling of two different oxidation states with two different charge-transfer states within one macrocyclic scaffold delivers up to five different optical outputs, where each switching node is characterized by a unique electronic absorptive signature based on oxidation state and local acidity. As shown in Fig. 5, **2** can be photochemically (or chemically) oxidized to rctt-**2**-[Ox$_1$] (without varying its conformation) then further chemically oxidized to rccc-**2**-[Ox$_2$] (with switching of conformer structure). rctt-**2**, rctt-**2**-[Ox$_1$] and rccc-**2**-[Ox$_2$] each have different optical properties (see Figs. 3, 4) and this stepwise oxidation process toggles between those states. Reduction of rctt-**2**-[Ox$_1$] can be achieved either electrochemically or chemically using Zn/acetic acid. However, on a larger scale, while these processes are essentially reversible, the latter is inconvenient due to a necessity for strongly acidic reagents, which are not compatible with macrocycle stability. Weaker acids may however be applied resulting in

the appearance of C-T bands with characteristic colors of deep purple for rctt-**2**-[Ox$_1$] and deep turquoise for rccc-**2**-[Ox$_2$], while rctt-**2** does not respond to acid. Appearance of the C-T band is reversible and depends on controlling acidity (Fig. 4) and can be achieved in solution or solid states. C-T state can also be switched by treatment of rctt-**2**-[Ox$_1$] with DDQ under acidic conditions (see Supplementary Figure 13) Each species involved in this series of reactions exhibits different optical properties as shown in Fig. 5. While the hemiquinhydrone C-T phenomena are clearly and reproducibly reversible in this manifold, the reversibility is complicated by several factors. First, the rctt-**2**-[Ox$_1$] to rctt-**2** transformation is possible under acidic conditions although these are usually detrimental to macrocycle persistence. Mildly acid conditions do permit rctt-**2** to be reformed. Second, reduction of rccc-**2**-[Ox$_2$] is complicated by the identity of the product, i.e., rctt-**2**-[Ox$_1$] or rccc-**2**-[Ox$_1$]. According to our DFT calculations,

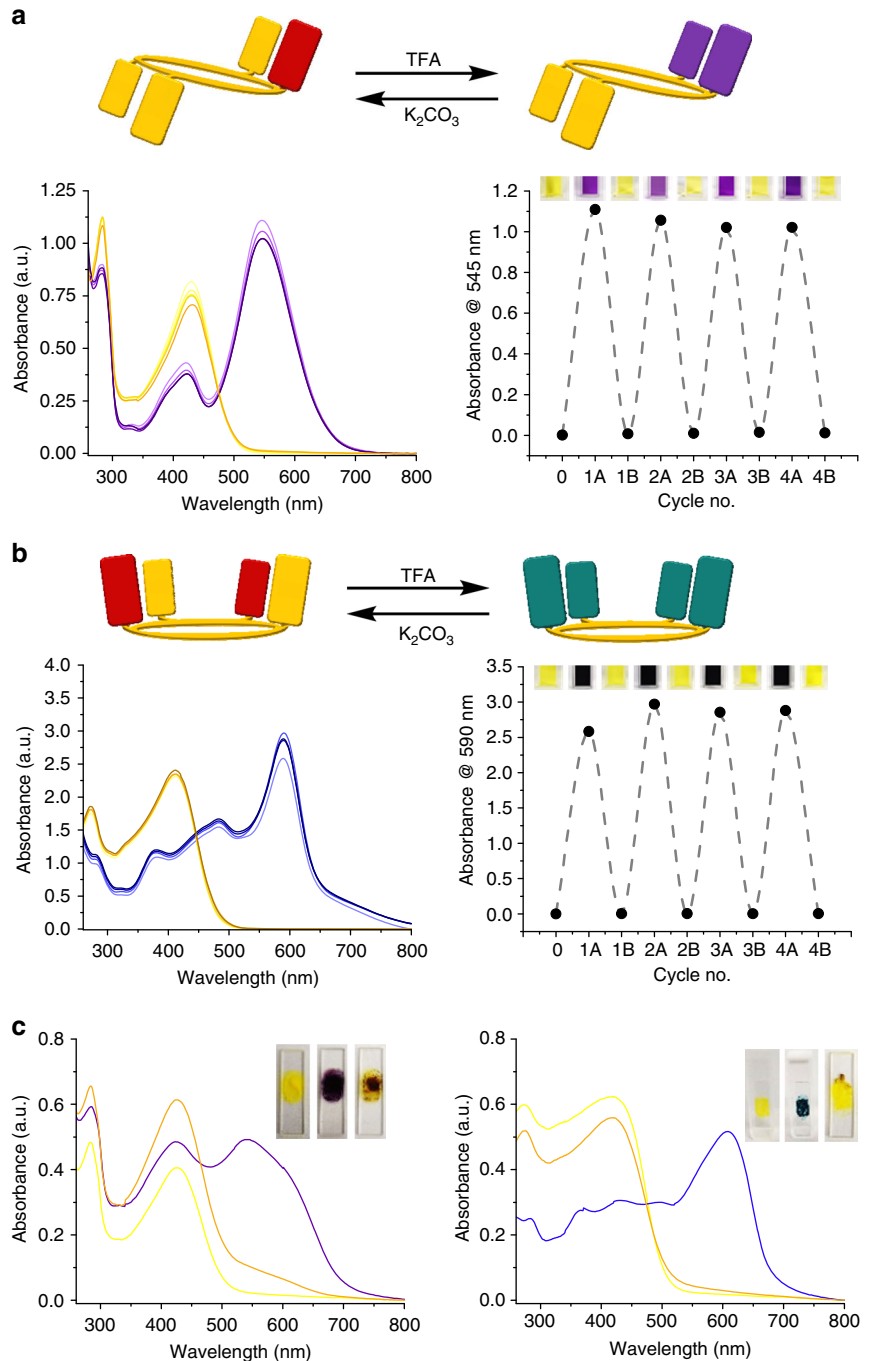

**Fig. 4** Charge-transfer state switching of rctt-**2**-[Ox$_1$] and rccc-**2**-[Ox$_2$]. **a** Top: graphical representation of hemiquinhydrone formation in rctt-**2**-[Ox$_1$] with (at left) the corresponding electronic absorption spectra of rctt-**2**-[Ox$_1$] during sequential acidification (trifluoroacetic acid (TFA), 50 μL) and neutralization (K$_2$CO$_3$) at $9.02 \times 10^{-5}$ mol dm$^{-3}$ in CHCl$_3$ demonstrating reversibility of charge-transfer complex formation with the alternating absorbance intensity shown at right (colored squares are photographs of the corresponding solution in cuvette). **b** Top: graphical representation of hemiquinhydrone formation in rccc-**2**-[Ox$_2$] with (at left) the corresponding electronic absorption spectra of rccc-**2**-[Ox$_2$] during sequential acidification (TFA, 175 μL) and neutralization (K$_2$CO$_3$) at $9.02 \times 10^{-5}$ mol dm$^{-3}$ in CHCl$_3$ and (at right) alternating absorbance intensity for the repeated process (colored squares are photographs of the corresponding solution in cuvette). **c** Solid-state electronic absorption spectra of rctt-**2**-[Ox$_1$] (left) and rccc-**2**-[Ox$_2$] (right) before (yellow spectra), during (purple/blue spectra) and after (orange spectra) exposure to TFA vapor. Inset photographs show the corresponding color changes in the solid-state samples deposited on quartz plate

rccc-**2**-[Ox$_1$] ought to be the main product, although this has proved difficult to determine experimentally.

## Discussion

A key feature of this switching manifold is the introduction of C-T complexes based on hemiquinones. Quinhydrone-type charge-

transfer complexes have conspicuous colors in solution and in solid mixtures of the compounds with acids including trifluoroacetic acid or *p*-toluenesulfonic acid (PTSA) because hydrogen bonding stabilizes quinhydrone formation. Hydroquinone and benzoquinone form a deep purple C-T complex by simple grinding (see Supplementary Figure 14; crystals of the

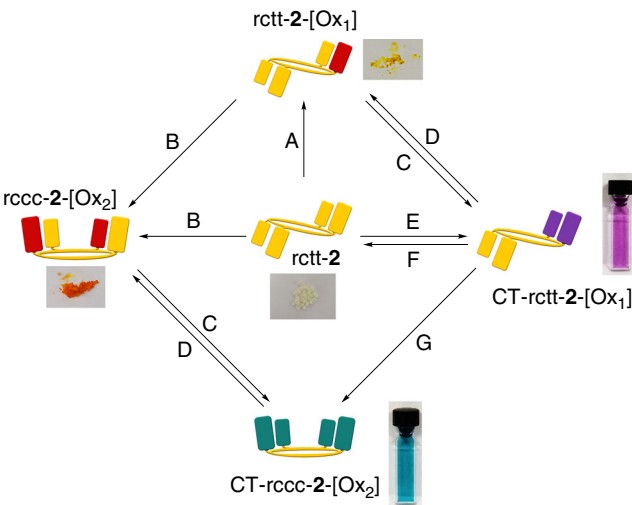

**Fig. 5** Chemical switching manifold based on **2**. Conditions (CHCl$_3$ solution): A: 2,3-dichloro-5,6-dicyano-1,4-benzoquinone (1 eq.) or irradiation (285 nm); B: 2,3-dichloro-5,6-dicyano-1,4-benzoquinone (4 equiv.); C: trifluoroacetic acid; D: potassium carbonate or pyridine; E: trifluoroacetic acid/irradiation (285 nm); F: Zn metal/acetic acid/90 °C; G: 2,3-dichloro-5,6-dicyano-1,4-benzoquinone/trifluoroacetic acid (see Supplementary Figure 13)

complex are green). Calix[4]arenequinones[25] were also eventually found to form quinhydrone-type complexes possessing a broad electronic absorption characteristic of C-T in their electronic absorption spectra[26,27]. However, the physical properties of C-T interactions between phenols and hemiquinones have not been previously reported, and therefore we set about determining the origin of the observation in the series **2** compounds. Both rctt-**2**-[Ox$_1$] and rccc-**2**-[Ox$_2$] exhibit the intense colors and characteristic C-T bands similar to quinhydrone complexes and are also ESR active as expected in the presence of organic acids (see Fig. 6 and Supplementary Figures 23–25 for compounds **2**). Formation of quinhydrone-type complexes is also supported by infrared spectroscopy (see Supplementary Figures 15,16).

Figure 6a shows the ultraviolet/visible (UV/vis) spectra of the **2** series and photographs of their solutions in the presence of TFA, revealing the hue of their quinhydrone charge-transfer complexes and characteristic absorption bands at 550 nm (rctt-**2**-[Ox$_1$]) and 600 nm (rccc-**2**-[Ox$_2$]). Quinhydrones contain unpaired electrons leading to characteristic ESR signatures depending on their structures. Figure 6b shows the corresponding ESR spectra taken from each compound mixed with PTSA in the solid state (solutions of quinhydrone complexes are often ESR silent due to molecular vibrational processes, as reported previously[28]). It was found that solid samples of series **2** in neutral state were weakly ESR active due to trace amounts of phenoxyl radical (Fig. 6c), which may not be involved in C-T complex formation under neutral conditions. Interestingly, while the phenoxyl radical spectrum is unaffected in the solid-state sample of rctt-**2** under acidic conditions, ESR spectra of rctt-**2**-[Ox$_1$] and rccc-**2**-[Ox$_2$] are broadened due to formation of the C-T state (due to delocalization of the radical). Quinhydrone complexes are known to be stabilized by hydrogen bonding with an acid such as TFA or PTSA (protonation of hemiquinone moieties makes them good electron acceptors which consequently interact with phenol groups by receiving electron density[29]). Thus, for rctt-**2**-[Ox$_1$], quinhydrone formation was only found in solvents of low polarity in the presence of an acid (Supplementary Figure 12). Although phenols are also known to form quinhydrone complexes with benzoquinones[30], the cases for compounds **2** are the first

examples of phenol/hemiquinone or hemiquinhydrone C-T complexes. An important feature of the presence of the radicals (albeit in low yield) in the neutral state of rccc-**2**-[Ox$_2$] is that equilibration of the substituents ought to occur to the lowest energy state by intramolecular electron/proton transfer processes.

For series **2** compounds, density functional theory (DFT) was used to investigate the origin of the two following unusual structural features: (a) an oxidation-induced conformational transformation from rctt to rccc, and (b) the observation of only a single isomer of the [Ox$_2$]-type products where 6 (including all rctt and rccc isomers) are possible despite there being no conjugation between *meso*-substituents in [Ox$_1$]-type compounds. The results are shown in Supplementary Figures 17 and 18 with Cartesian coordinates of the calculated structures given in Supplementary Tables 1–8. Substituent geometries (from X-ray crystal structures) at the *meso*-positions approach tetrahedral for singly bonded substituents (i.e., phenols) but are approximately trigonal for oxidized hemiquinonoid substituents (see Supplementary Figure 19). For rctt-**2**-[Ox$_1$], a single hemiquinonoid subsituent can be accommodated without significant variation in the rctt structure. In fact, in the crystal structure of rctt-**2**-[Ox$_1$] the quinone site is delocalized over two sites. Oxidation of rctt-**2**-[Ox$_1$] gives exclusively rccc-**2**-[Ox$_2$] by an unusual conformational switching of resorcinarenes. According to DFT calculations, rccc-**2**-[Ox$_2$] is similar in stability to rctt-**2**-[Ox$_2$] with only an approximately 2.3 kcal mol$^{-1}$ difference in total energies (rccc-**2**-[Ox$_1$] and rctt-**2**-[Ox$_1$] also have similar stabilities; see Supplementary Figures 17,18). The origin of the rctt–rccc transformation[15,31] probably lies in the more easy accommodation of two unsaturated *meso*-positions in the rccc form. That the rccc isomer of **2**-[Ox$_2$] is favored might also suggest that (a) there is a low barrier for the rctt–rccc transformation in rctt-[Ox$_1$] and (b) the rccc-**2**-[Ox$_1$] form is more easily oxidized than its rctt isomer under the conditions used. Regardless of the route, the single isomer 'trans'-rccc-**2**-[Ox$_2$] is the sole product of the process. This is allowed by equilibration of substituents caused by redox interactions involving intramolecular quinhydrone formation or, in this case, hemiquinhydrone since it involves hemiquinonoid and phenol substituents. Intermolecular exchange by this mechanism is known[7,8] and has recently been reported for intramolecular substituent exchange in chiral benzoquinone/hydroquinone compounds[28]. For the **2** system, this explains the existence of a single 'trans'-rccc-**2**-[Ox$_2$] isomer since both 'cis'-rccc-**2**-[Ox$_2$] and 'gem'-rccc-**2**-[Ox$_2$] isomers are less stable (by 6.3 kcal mol$^{-1}$ and 20.5 kcal mol$^{-1}$, respectively; see Supplementary Figures 17,18).

In summary, we report charge-transfer switching involving antioxidant-substituted resorcinarene macrocycles whose properties are based on oxidation-induced conformational changes, photochemical switching and hemiquinhydrone formation. The multiplexing of these properties has allowed us to construct an optical switching system based on oxidation state variation and the corresponding differences between the C-T complexes of the different oxidation states. These are based on rather subtle intramolecular processes, although changes in optical properties are intense. Oxidation state variation consecutively from rctt-**2** to rctt-**2**-[Ox$_1$] to rccc-**2**-[Ox$_2$] leads to ratiometric differences in their main absorbance bands while each of the compounds exhibits different hemiquinhydrone complex formation behavior respectively from none to single to double. This molecular switching manifold couples two different oxidation states with two different charge-transfer states within one macrocyclic scaffold delivering up to five different optical outputs, and exploits intramolecular coupling of multiple redox-active substituents within a single molecule. The photoresponsivity and multistability of these molecules make them suitable for molecular

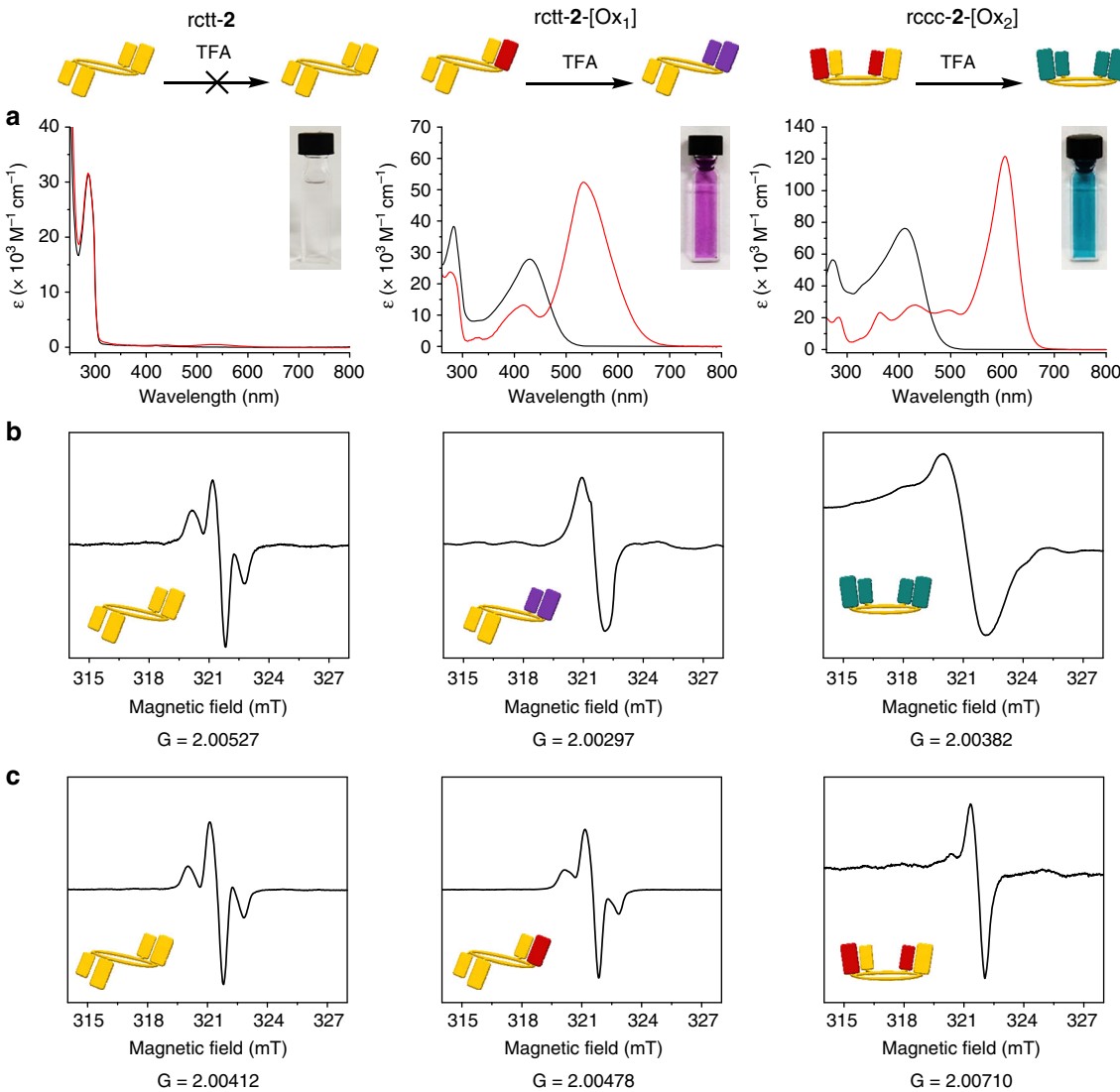

**Fig. 6** Electronic absorption and X-band electron spin resonance (ESR) spectra of series **2** compounds. **a** Electronic absorption spectra of rctt-**2** (left), rctt-**2**-[Ox₁] (middle) and rccc-**2**-[Ox₂] (right) in neutral (black lines, CHCl₃, [rctt-**2**], [rctt-**2**-[Ox₁]] and [rccc-**2**-[Ox₂]] = 2.47 × 10⁻⁵ mol) and acidified (red lines, [rctt-**2**] = 1.02 × 10⁻³ mol dm⁻³ in TFA/CHCl₃ (89 mM), [rctt-**2**-[Ox₁]] and [rccc-**2**-[Ox₂]] = 4.94 × 10⁻⁵ mol dm⁻³ in TFA) solutions. Insets show the appearance of the acidified solutions (see also Supplementary Figure 12). **b** ESR spectra of rctt-**2** (left), rctt-**2**-[Ox₁] (middle) and rccc-**2**-[Ox₂] (right) in *p*-toluenesulfonic acid in the solid state (acidified solutions of the compounds are ESR silent). **c** ESR spectra rctt-**2** (left), rctt-**2**-[Ox₁] (middle) and rccc-**2**-[Ox₂] (right)

logic gate operations, molecular switches, chemical memory elements or chemosensors.

## Methods

**General information**. Reagents and dehydrated solvents (in septum-sealed bottles) used for syntheses and spectroscopic measurements were obtained from Tokyo Kasei Chemical Co., Wako Chemical Co. or Aldrich Chemical Co. and were used without further purification. Details of the synthesis of the compounds are given in Supplementary Methods. Electronic absorption spectra were measured using JASCO V-570 UV/Vis/NIR spectrophotometer, Princeton Applied Research (PAR) diode array rapid scanning spectrometer or a Shimadzu UV/Visible spectrophotometer. Fluorescence spectra were measured using a JASCO FP-670 spectrofluorimeter. Fourier-transform infrared spectroscopy (FTIR) spectra were obtained from solid samples using a Thermo-Nicolet 760X FTIR spectrophotometer. ESR spectra were measured from solid samples using a JEOL JES-FA200 spectrometer with data recorded and processed using A-System version 1.6.5 PCI J/X-Band and FA-Manager version 1.2.9 V2 series. ¹H nuclear magnetic resonance (NMR) spectra were recorded on a JEOL AL300BX NMR spectrometer at 300 MHz, and proton decoupled ¹³C NMR were recorded at 75 MHz on a JEOL AL300BX NMR spectrometer at the stated temperatures. Data were processed on Delta version 5.0.5.1 and Always JNM-AL version 6.2. ¹H NMR chemical shifts are reported in ppm relative to tetramethylsilane for CDCl₃ (δ 0.00) or the residual solvent peak for other solvents. ¹³C NMR chemical shifts are reported in ppm relative to the

solvent reported. Coupling constants (*J*) are expressed in Hertz (Hz), shift multiplicities are reported as singlet (s), doublet (d), triplet (t), quartet (q), double doublet (dd), multiplet (m) and broad singlet (bs). High-resolution electrospray ionization-mass spectrometry measurements were performed using the Waters Ltd. Xevo-G2-XS-ToF spectrometer with direct infusion of samples using an Alliance e2695 HPLC system. Samples were dissolved in methanol. Electrospray conditions: cone voltage: 40 V; capillary voltage: 3 kV; desolvation temperature: 350 °C; desolvation gas flow (dry nitrogen, 500 L h⁻¹); source temperature (120 °C). Leucine enkephalin was used as internal reference for high-resolution mass spectrometry (for positive ion: molecular weight (M.W.) = 556.2771 Da; for negative ion: M.W. = 554.2615 Da).

**Analytical data**. The ¹H and ¹³C NMR spectra and high-resolution mass spectra of the compounds are shown in Supplementary Figures 26–37.

**Electrochemistry**. Cyclic voltammograms were recorded on an EG&G Model 263 A potentiostat using a three-electrode system. A platinum button electrode was used as the working electrode. A platinum wire served as the counter electrode and an Ag/AgCl electrode was used as the reference. Ferrocene/ferrocenium redox couple was used as an internal standard. All the solutions were purged prior to electrochemical and spectral measurements using argon gas. Spectroelectrochemical study was performed using a cell assembly (SEC-C) supplied by ALS Co., Ltd. (Tokyo, Japan). This assembly comprised a Pt counter electrode, a 6 mm Pt gauze working electrode and

an Ag/AgCl reference electrode in a 1.0 mm path length quartz cell. The optical transmission was limited to 6 mm covering the Pt gauze working electrode.

**X-ray crystallography**. Crystals of rctt-**1** were grown by layering a solution of rctt-**1** in dimethylsulfoxide with acetone. Pale yellow rhombs grew within several hours. Crystals of the other compounds were grown by diffusion of hexane into solutions of the compounds in chloroform. Data collections were performed using MoK$_\alpha$ radiation ($\lambda = 0.71073$ Å) on a RIGAKU VariMax Saturn diffractometer equipped with a charge-coupled device (CCD) detector or a Bruker APEX CCD diffractometer. Prior to the diffraction experiment the crystals were flash-cooled to the given temperatures in a stream of cold N$_2$ gas. Cell refinements and data reductions were carried out using the d*trek program package in the CrystalClear software suite[32]. The structures were solved using a dual-space algorithm method (SHELXT)[33] and refined by full-matrix least squares on F2 using SHELXL-2014[34] in the WinGX program package[35]. Non-hydrogen atoms were anisotropically refined and hydrogen atoms were placed on calculated positions with temperature factors fixed at 1.2 times $U_{eq}$ of the parent atoms and 1.5 times $U_{eq}$ for methyl groups. Crystal data for rctt-**1**: pale yellow block, C$_{108}$H$_{180}$O$_{26}$S$_{12}$, M$_r$ = 2279.23, triclinic P-1, a = 13.2004(12) Å, b = 15.5166(13) Å, c = 15.8355(12) Å, α = 72.51(2)°, β = 81.085 (10)°, Υ = 78.12(2)°, V = 3051.0(6) Å$^3$, T = 123 K, Z = 1, R$_{int}$ = 0.0664, GoF = 1.060, R$_1$ = 0.0921, wR(all data) = 0.2973. Crystal data for rctt-**2**: colorless bar, C$_{72.18}$H$_{81.08}$O$_6$, M$_r$ = 1044.56, triclinic P-1, a = 13.3000(13) Å, b = 14.8978(14) Å, c = 16.5911(16) Å, α = 82.520(2)°, β = 79.599(2)°, Υ = 78.563(2)°, V = 3153.9(5) Å$^3$, T = 160 K, Z = 2, R$_{int}$ = 0.0441, GoF = 1.245, R$_1$ = 0.1134, wR(all data) = 0.3688. Crystal data for rctt-**2**-[Ox$_1$]: yellow block, C$_{157}$H$_{169}$O$_{13}$, M$_r$ = 2264.92, triclinic P-1, a = 13.4663(14) Å, b = 15.0165(16) Å, c = 16.4398(17) Å, α = 88.39(2)°, β = 80.62 (2)°, Υ = 81.48(2)°, V = 3243.7(6) Å$^3$, T = 123 K, Z = 1, R$_{int}$ = 0.1107, GoF = 0.989, R$_1$ = 0.0703, wR(all data) = 0.2260. Crystal data for rccc-**2**-[Ox$_2$]: orange bar, C$_{150}$H$_{160}$O$_{12}$, M$_r$ = 2154.77, monoclinic C 2/c, a = 35.8013(17) Å, b = 16.1205(11) Å, c = 25.9004(15) Å, β = 123.408(8)°, V = 12478.2(87) Å$^3$, T = 293 K, Z = 4, R$_{int}$ = 0.0675, GoF = 1.019, R$_1$ = 0.673, wR(all data) = 0.2477.

**Computational methods**. All structures presented in this work were fully optimized using the DFT ωB97XD/4-31G functional[36] with localized 6-31G basis set as it is implemented in the program Gaussian 09[37]. The applied exchange-correlation functional includes empirical correction for dispersion energy that is important for correct description of host–guest interactions. Effects of the solvent environment, chloroform in the present case, was simulated by implicit IEFPCM model[38–40] with ε parameter set to 4.7113. Cartesian coordinates of the calculated structures are given in Supplementary Tables 1–8.

**Demonstration of charge-transfer state switching**. Solution switching studies shown in Fig. 4 were recorded using a 1 cm quartz cell with 4 polished faces at room temperature. TFA was added directly to the resorcinarene solution in the cuvette using a volumetric pipette and the electronic absorption spectrum was measured. After measurement, the acidic solution was neutralized by passing it through a plug of potassium carbonate leading to a solution having a color the same as the starting resorcinarene solution. Note that organic bases could not be used for neutralization since salt formation in the chlorinated solvent leads to formation of an emulsion in the cuvette, which prevented measurement of its electronic absorption spectrum. Solid-state switching studies were made using a quartz plate at room temperature. Thin films of the compounds were prepared by drop-casting of a dichloromethane solutions of the respective compound directly on the plate. TFA vapor was collected from the headspace of the open TFA receptacle using a Pasteur pipette and the contents were passed across the surface of the film until a color change was obtained. Alternatively, the plate was placed in a quartz cuvette filled with TFA vapor and the electronic absorption spectrum recorded in situ.

## Data availability
Data supporting the findings of this study are available in the Supplementary Information. The Supplementary Information contains full details on the synthesis and characterization of compounds, switching studies and further details of computational studies. X-ray crystallographic data have been deposited at Cambridge Crystallographic Data Centre (CCDC) under deposition numbers 1867498, 1857780, 1867500, and 1858690. These data can be obtained free of charge from The Cambridge Crystallographic Data Centre via www.ccdc.cam.ac.uk/data_request/cif.

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

## Acknowledgements

This work was partly supported by World Premier International Research Center Initiative (WPI Initiative), MEXT, Japan. The authors are grateful to Japan Society for the Promotion of Science (JSPS) for a JSPS Fellowship (to D.T.P.). This work was also partially supported by JSPS KAKENHI (Coordination Asymmetry) (Grant No. JP16H06518), and CREST, JST (Grant No. JPMJCR1665). This work was partly financially supported by the National Science Foundation (Grant No. 1401188 to F.D.). W.S. and N.Z. thank the Science Foundation Ireland (SFI; 13/IA/1896) and the European Research Council (CoG 2014-647719) for financial support. D.T.P. thanks the University of Birmingham for a PhD studentship and the National Institute for Materials Science (NIMS, Japan) for a NIMS internship under which preliminary investigations for this work were made. D.T.P. also thanks The Society for Chemical Industry (SCI, UK) for a Messel Bursary which facilitated initial research exchange for this work and the Royal Society of Chemistry (RSC, UK) for a conference travel grant. The authors are grateful to Chi Tsang (University of Birmingham) for assistance with mass spectrometric analyses. The authors thank the Catalysis and Sensing for our Environment (CASE) network for essential networking opportunities.

## Author contributions

D.T.P. and J.P.H. designed and performed the synthesis and switching experiments. Y.M., N.Z. and W.S. determined the X-ray crystal structures. W.A.W. and F.D. carried out electrochemical analysis. J.L. undertook spectroscopic measurements. Z.F. carried out DFT calculations. W.J. and N.F. aided with the collection of ESR data. J.S.F., K.A. and J.P. H. analyzed spectroscopic data and directed the research. All authors contributed to discussions throughout the project and the final writing and editing of the manuscript.

## Additional information

**Competing interests:** The authors declare no competing interests.

**Journal Peer Review Information:** *Nature Communications* thanks Giuseppe Trusso Sfrazzetto and other anonymous reviewer(s) for their contribution to the peer review of this work. Peer reviewer reports are available.

