## [Peer Review File · Nature Communications]

Reviewers' comments:

Reviewer #1 (Remarks to the Author):

The authors have undertaken an extremely interesting study into the use of non-cone conformer resorcinarene to generate a switching manifold. As the authors correctly state, much of the supramolecular chemistry surrounding Res[4]s has focused on forming capsules and studying host-guest chemistry of these systems (primarily in solution). The current work is a departure from that area and delves into the difficult topic of bridge modification and application / utility. This is not straightforward chemistry and the authors have done an excellent job in elucidating the behaviour of this system. The paper is very well written and I have no suggestions for improvement - I didn't even spot a typo, thank you. The figures are excellent and guide the reader perfectly through the story. Overall this is an outstanding piece of work that should be published in Nat Comms as is. I have checked over the SI and XRD data. I am not concerned by the CIF alerts as these are typical for systems such as this, especially when they contain disordered solvent / molecules / fragments etc. The DFT work could be checked by a specialist in the area, but all looks good to me. This paper was a pleasure to read.

Reviewer #2 (Remarks to the Author):

Manuscript of Hill and co-workers reports of the synthesis and characterization of a new resorcinarene, able to switch into 5 different molecular entities, by using a multimodal chemical or photochemical oxidative process. In particular, the introduction of a hemiquinhydrone moiety on the macrocyclic scaffold leads to the possibility to obtain a chemical species sensitive to pH and oxidation/reduction processes.

The compounds obtained are well characterized, the experiments carried out in order to validate and demonstrate the multimodal switching are well designed and performed. In addition, English is easily understandable.

Today, this kind of molecular switches are very important, due to the wide applications. Thus, after some minor revisions of the manuscript (reported below), I suggest the publication in Nature Communications.

Revisions:

- References 6 and 7, reported in the caption of Figure 1, should be moved into the main text;
- Resorcinarenes are characterized by the formation of an intramolecular hydrogen bond network, crucial to guarantee the "cone" conformation of the host, between OH groups on the upper rim. The formation of this network is also the driving force that leads to high yields in the resorcinarene synthesis. In this work, the derivatives are synthesized starting from an aromatic aldehyde bearing tert-butyl groups, thus leading to a macrocycle without the possibility to form this H-bond network. Compound 1 is obtained as single isomer with 75% of yield. How is it possible, considering the impossibility to lead the H-bond network on the upper rim? This point should be mentioned by the authors in the manuscript.
- Figure 2d is not cited in the manuscript at page 6. But, it can be merged with figure 3.
- Page 7: the authors describe: "rctt-2-[Ox1] has its hemiquinonoid substituent delocalized over two

diagonally-opposing sites in its X-ray crystal structure...". Is it correct? Or they are referred to rccc-2-[Ox2]?

- Some information about the stability of the compounds after the oxidation process are required.
- The nature of the band at 600 nm in compound rctt-2-[Ox1] is not clear for me: in the caption of Figure 3, it is to ascribe to the formation of an acid-stabilized C-T complex, while in the main text it is reported to the formation of HCl or singlet oxygen (I suppose that the singlet oxygen cannot be followed by simple UV measurements)? The authors are invited to clarify the origin of this new band in the main text.
- Figure 3c shows that rctt-2-[Ox1] can be obtained by photooxidation of rctt-2 in TFA/CHCl₃, followed by neutralization by addition of pyridine; however, in the main text (page 9), is reported that "This process can be operated in reverse by applying Zn metal in acetic acid for reduction back to rctt-2 albeit in low yield due to the instability of rctt-2 in acidic media". What is the correct procedure?
- In solid phase (Figure 4c), the recovery of the rctt-2-[Ox1] structure after the exposure of TFA vapors occurs without the presence of a base. This point, if correct, should be highlighted by the authors in the main text.
- In the acid/base cycles (Figure 4), the use of K₂CO₃ as base in CHCl₃ can lead to a non-homogeneous solution, precluding a correct UV-Vis measurement. I suggest the employ of triethylamine.
- In the conclusion section, some application of this multimodal molecular switch can be described.

Responses to the Reviewers' comments:

Reviewer #1 (Remarks to the Author):

The authors have undertaken an extremely interesting study into the use of non-cone conformer resorcinarene to generate a switching manifold. As the authors correctly state, much of the supramolecular chemistry surrounding Res[4]s has focused on forming capsules and studying host-guest chemistry of these systems (primarily in solution). The current work is a departure from that area and delves into the difficult topic of bridge modification and application / utility. This is not straightforward chemistry and the authors have done an excellent job in elucidating the behaviour of this system. The paper is very well written and I have no suggestions for improvement - I didn't even spot a typo, thank you. The figures are excellent and guide the reader perfectly through the story. Overall this is an outstanding piece of work that should be published in Nat Comms as is. I have checked over the SI and XRD data. I am not concerned by the CheckCIF alerts as these are typical for systems such as this, especially when they contain disordered solvent / molecules / fragments etc. The DFT work could be checked by a specialist in the area, but all looks good to me. This paper was a pleasure to read.

Response:

We would like to thank the reviewer for the positive comments about our manuscript. We are also grateful for the helpful comments regarding the CheckCIF files and the confirmation that these crystallographic alerts are common for these systems.

Reviewer #2 (Remarks to the Author):

Manuscript of Hill and co-workers reports of the synthesis and characterization of a new resorcinarene, able to switch into 5 different molecular entities, by using a multimodal chemical or photochemical oxidative process. In particular, the introduction of a hemiquinhydrone moiety on the macrocyclic scaffold leads to the possibility to obtain a chemical species sensitive to pH and oxidation/reduction processes.

The compounds obtained are well characterized, the experiments carried out in order to validate and demonstrate the multimodal switching are well designed and performed. In addition, English is easily understandable.

Today, this kind of molecular switches are very important, due to the wide applications. Thus, after some minor revisions of the manuscript (reported below), I suggest the publication in Nature Communications.

Response:

We would like to thank the reviewer for their positive comments regarding the scientific rigor of the studies and for providing constructive revisions to the manuscript. We have

included below responses to the reviewer's comments and changes made in the manuscript.

Revisions:

References 6 and 7, reported in the caption of Figure 1, should be moved into the main text;

Response:

We have moved the References 6 and 7 into the main text as requested (Pg. 3, Ln. 9)

Resorcinarenes are characterized by the formation of an intramolecular hydrogen bond network, crucial to guarantee the 'cone' conformation of the host, between OH groups on the upper rim. The formation of this network is also the driving force that leads to high yields in the resorcinarene synthesis. In this work, the derivatives are synthesized starting from an aromatic aldehyde bearing tert-butyl groups, thus leading to a macrocycle without the possibility to form this H-bond network. Compound 1 is obtained as single isomer with 75% of yield. How is it possible, considering the impossibility to lead the H-bond network on the upper rim? This point should be mentioned by the authors in the manuscript.

Response:

Studies reported by others working in the field of resorcinarenes have highlighted preference for formation of the *rcct* 'chair' isomer in cases where aryl aldehydes are used in the synthesis rather than alkyl aldehydes (alkyl aldehydes generally prefer the *rccc* 'cone' isomer mentioned by the reviewer). Previous work indicates that steric requirements are also important in determining the major isomer formed. In this case, the bulky di-*t*-butyl-hydroxyphenyl groups promote formation of the *rcct* isomer where steric hindrance is minimized similarly to the case for other bulky aromatic aldehydes¹.

The factors affecting major product isomer identity are discussed in a review by Reinhoudt and co-workers², who gave an in-depth analysis of the multiple driving forces involved in isomer selection. The existence of non-cone isomers has been known since preliminary work by Högberg³.

References:

1. P. Sakhaii, I. Neda, M. Freytag, H. Thönnessen, P. G. Jones, R. Schmutzler, Stereoselective synthesis and structure of new types of calix[4]resorcinarenes. Complexation of tetrakis(O,O-Phosphorus)-bridged-calix[4]resorcinarenes with heavy metal atoms. *Z. Anorg. Allg. Chem.* 2000, 626, 1246–1254.
2. P. Timmerman, W. Verboom, D. N. Reinhoudt, Resorcinarenes. *Tetrahedron*, 1996, 52, 2663–2704.

3. A. G. S. Högberg, Two stereoisomeric macrocyclic resorcinol-acetaldehyde condensation products, *J. Org. Chem.* 1980, 45, 4498–4500.

Action:

The following statement (with accompanying references added as citations 22, 23) has been added to the manuscript to account for this point:

“The exclusive formation of *rctt-1* is consistent with previous reports for the synthesis of resorcinarenes from aryl aldehydes^{22,23}. (Pg. 5, Ln. 11)

Figure 2d is not cited in the manuscript at page 6. But, it can be merged with figure 3.

Response:

We have now referenced Figure 2d in the main text and highlighted its link with Figure 2c.

Action:

We now mention Figure 2d in the main text including indicating that it is a graphical representation of Figure 2c. Pg. 5, Ln 17.

We have also added reference for Figure 2b, which was missing “(*rctt*, for other isomer forms see Figure 2b)” Pg. 5, Ln. 9 of main text.

Page 7: the authors describe: *rctt-2*-[Ox1] has its hemiquinonoid substituent delocalized over two diagonally-opposing sites in its X-ray crystal structure. Is it correct? Or they are referred to *rccc-2*-[Ox2]?

Response:

It is correct – we are referring to *rctt-2*-[Ox₁]. To clarify this point, in the X-ray crystal structure there are two possible isostructural positions where the carbonyl could reside. This leads to the effect of observing the carbonyl delocalized over these two positions, causing the observed bond length for these two C-O bonds to be the average of that expected for one each of single C-O and double C=O bond. However, in solution one carbonyl type ¹³C NMR resonance is observed for *rctt-2*-[Ox₁] while *rctt-2*-[Ox₂] possesses two carbonyl resonances clearly differentiating the two compounds.

Action:

We have modified the sentence structure to differentiate the oxidized compounds more clearly. The following detail was added in Pg 5:

...although in solution it is clearly localized at one site. Pg 5, last line

Some information about the stability of the compounds after the oxidation process are required.

Response:

The oxidized compounds are stable in the solid state over periods of several years. In neutral solutions, the oxidized compounds show no observable degradation neither during extended spectroscopic measurements nor while under crystallization for X-ray measurements. Resorcinarenes are not stable in strongly acidic solutions although under the conditions we have used here (trifluoroacetic acid) to demonstrate switching no degradation was observed over periods of several hours.

Action:

Beyond what we have already mentioned about the stability (Pg 12 of the original submission), we have added the following statements regarding the stabilities of the oxidized compounds:

“The oxidized compounds are stable in the solid state over periods of several years. In neutral solutions, the oxidized compounds show no observable degradation either during extended spectroscopic measurements or while under crystallization for X-ray measurements.” Pg 6, Ln. 3-6:

The nature of the band at 600 nm in compound *rctt-2*-[Ox₁] is not clear for me: in the caption of Figure 3, it is to ascribe to the formation of an acid-stabilized C-T complex, while in the main text it is reported to the formation of HCl or singlet oxygen (I suppose that the singlet oxygen cannot be followed by simple UV measurements)? The authors are invited to clarify the origin of this new band in the main text.

Response:

In this example, two different processes are occurring simultaneously. Irradiation of the sample leads to the photo-oxidation of *rctt-2* to *rctt-2*-[Ox₁]. Formation of singlet oxygen cannot be monitored using UV/vis and requires a technique such as ESR spectroscopy for its detection (see Fig. S10). Due to the short wavelength light required for the photo-oxidation, the solvent (CHCl₃) is decomposed leading to the presence of HCl in solution, which in turn leads to the formation of the charge transfer complex in observable quantities during the experiment (signified by the appearance of the 600 nm band). If the photo-oxidation is carried out in the presence of a solid base (such as K₂CO₃) then this is avoided and only the oxidation bands are observed in the UV-Vis spectra, however the presence of a solid base during the recording of the UV-Vis time course experiments led to significant light scattering and was therefore omitted in the final experiments.

Action:

Figure 3c caption has been revised to include a statement to account for the origin of the 600 nm band. This effect was already briefly mentioned in the text on Pg. 8 of the original submission.

“...formed as HCl is generated in situ by the UV-promoted decomposition of CHCl_3 .”
Figure 3 caption.

Figure 3c shows that *rctt-2*-[Ox₁] can be obtained by photooxidation of *rctt-2* in TFA/ CHCl_3 , followed by neutralization by addition of pyridine; however, in the main text (page 9), is reported that this process can be operated in reverse by applying Zn metal in acetic acid for reduction back to *rctt-2* albeit in low yield due to the instability of *rctt-2* in acidic media. What is the correct procedure?

Response:

These are actually two different processes. The addition of pyridine is not a reduction process but a neutralization of the acid responsible for the formation of the charge transfer complex of *rctt-2*-[Ox₁], which is formed directly after the photochemical oxidation of *rctt-2* in TFA/ CHCl_3 . Acid neutralization leads to formation of the non-charge transfer *rctt-2*-[Ox₁] species in neutral solution. On the other hand, Zn metal in acetic acid can be used to reduce *rctt-2*-[Ox₁] from its charge transfer complex *rctt-2*-[Ox₁] to *rctt-2*.

Action:

Pg7. A statement has been added to clarify the role of the pyridine in the neutralization of acid:

“Neutralization of the resulting deep purple solution with pyridine gives a yellow solution with an electronic absorption spectrum characteristic of *rctt-2*-[Ox₁].” Pg7, Ln. 13-14.

In solid phase (Figure 4c), the recovery of the *rctt-2*-[Ox₁] structure after the exposure of TFA vapors occurs without the presence of a base. This point, if correct, should be highlighted by the authors in the main text.

Response:

This point has now been highlighted in the main text of the manuscript.

Action:

The following statement was added to highlight this point:

“In the solid state, if a volatile acid such as TFA is employed for C-T formation, then neutralization with base is not required since evaporation of the acid returns the molecule

to its original state.” Pg. 8 Ln. 1.

In the acid/base cycles (Figure 4), the use of K_2CO_3 as base in $CHCl_3$ can lead to a non-homogeneous solution, precluding a correct UV-Vis measurement. I suggest the employ of triethylamine.

Response:

Solutions were passed through a potassium carbonate plug prior to each UV/vis measurement. We have attempted to use an organic base in situ for this process. However, the solution becomes turbid due to formation of an emulsion caused by the formation of the salt of the organic base, which prevents the recording of UV-Vis data. We have now included this caveat in the Supplementary Information.

Action:

A detailed description of the switching study has now been added to the Methods section including the reason why organic bases cannot be used for solution state operation.

In the conclusion section, some application of this multimodal molecular switch can be described.

Response:

Some potential applications have been included in the conclusion section.

Action:

We have added the following statement regarding the potential applications in the conclusion:

“The photoresponsivity and multi-stability of these molecules make them suitable for molecular logic gate operations, molecular switches, chemical memory elements or chemosensors.” Pg. 12, Ln. 16.

REVIEWERS' COMMENTS:

Reviewer #1 (Remarks to the Author):

Changes are absolutely fine, publish as is.

Reviewer #2 (Remarks to the Author):

The authors modified the manuscript according with my observations. Furthermore, I would thank the authors to have been clarified in the response letter some points, previously not clear for me. The manuscript, in my opinion, is now suitable for the publication in Nature Communication.

Giuseppe Trusso Sfrazzetto